# Design of highly efficient deep-blue organic afterglow through guest sensitization and matrices rigidification

Shen Xu [1,4], Wu Wang[1,4], Hui Li[1], Jingyu Zhang[1], Runfeng Chen [1✉], Shuang Wang[1], Chao Zheng[1], Guichuan Xing [2✉], Chunyuan Song [1] & Wei Huang [1,3✉]

Blue/deep-blue emission is crucial for organic optoelectronics but remains a formidable challenge in organic afterglow due to the difficulties in populating and stabilizing the high-energy triplet excited states. Here, a facile strategy to realize the efficient deep-blue organic afterglow is proposed via host molecules to sensitize the triplet exciton population of guest and water implement to suppress the non-radiative decays by matrices rigidification. A series of highly luminescent deep-blue (405–428 nm) organic afterglow materials with lifetimes up to 1.67 s and quantum yields of 46.1% are developed. With these high-performance water-responsive materials, lifetime-encrypted rewritable paper has been constructed for water-jet printing of high-resolution anti-counterfeiting patterns that can retain for a long time (>1 month) and be erased by dimethyl sulfoxide vapor in 15 min with high reversibility for many write/erase cycles. These results provide a foundation for the design of high-efficient blue/deep-blue organic afterglow and stimuli-responsive materials with remarkable applications.

[1] Key Laboratory for Organic Electronics and Information Displays & Jiangsu Key Laboratory for Biosensors, Institute of Advanced Materials (IAM), Jiangsu National Synergistic Innovation Center for Advanced Materials (SICAM), Nanjing University of Posts and Telecommunications, 9 Wenyuan Road, Nanjing 210023, People's Republic of China. [2] Institute of Applied Physics and Materials Engineering, University of Macau, Macao SAR 999078, China. [3] Shaanxi Institute of Flexible Electronics (SIFE), Northwestern Polytechnical University (NPU), 127 West Youyi Road, Xi'an 710072, People's Republic of China. [4] These authors contributed equally: Shen Xu, Wu Wang. ✉email: iamrfchen@njupt.edu.cn; gcxing@um.edu.mo; iamwhuang@nwpu.edu.cn

Blue luminescence, which has long been documented as a prime challenge in both fluorescent and phosphorescent materials[1,2], is the core elements indispensable for solid-state lighting and full-color display technologies in organic optoelectronics[3–5]. Organic ultralong room temperature phosphorescence (OURTP) with luminescent lifetime over 0.1 s after removing the excitation source has attracted considerable attention due to the fundamental breakthrough of the excited state lifetime tuning for organic afterglow with unique photophysical properties and innovative applications in many fields[6,7]. However, despite of the high molecular diversity of organic molecules, most organic afterglow emission bands are in a range from 500 to 600 nm owing to its triplet state and solid state luminescence nature; both of them will lead to low-lying exciton energies and bathochromic shifts of the emission spectrum[8,9]. It becomes even more challenging to develop blue OURTP, which should populate and stabilize the high-lying triplet excited state simultaneously[9,10].

Crystallization/H-aggregation[11,12] and exciplex formation[13–15] strategies previously proposed to construct OURTP are difficult to realize the blue organic afterglow owing to the unavoidable red-shift compared to the single-molecular fluorescence after the aggregation coupling and intermolecular electronic interaction in solid states[16]. Dispersing emitters in host is effective in preventing the bathochromic shift and eliminating the concentration quenching by inhibiting molecular aggregation and electronic coupling at low doping concentrations[17]. To achieve highly efficient OURTP, rigid host matrixes are essentially needed to suppress the non-radiative relaxation decays for high phosphorescent quantum yield (PhQY)[18,19]. Nevertheless, rigid host molecules with strong intermolecular interactions are generally hard to be processed and the single molecular dispersion of the dopant suffers from the low compatibility between the highly polar host and luminescent guest[20]. Moreover, the luminance of the host-guest material is usually weak due to the small amount of the doped emitters in the optically inert host. Therefore, despite the recent efforts[21,22], it remains difficult to achieve efficient blue/deep-blue organic afterglow with long lifetime (>1.0 s) and high PhQY (>40%).

Here, we propose a general strategy to overcome these intrinsic difficulties in designing blue organic afterglow using active host for triplet excited state sensitization and water implement for matrices rigidification to simultaneously elongate the lifetime and boost the PhQY of OURTP. Guest sensitization through efficient energy transfer from host to guest promotes significantly the excitation of the small amount (~0.5%) of doped guest molecules for strong OURTP. Matrices rigidification using water, inspired by the solidification process of concrete[23], greatly rigidifies the host-guest system by forming hydrogen-bonding (H-bonding) networks and this post-rigidification approach makes the rigid hosts not obligated, which should significantly facilitate the material design and preparation of high-performance organic afterglow materials. Indeed, a series of deep-blue organic afterglow materials are achieved and the OURTP lifetimes are improved up to 1.67 s and PhQY to 46.1%, which are among the best organic afterglow performance reported so far[7,8]. Further, in light of the extraordinary water-responsible OURTP, rewritable lifetime-encryption paper for anti-counterfeiting is constructed. High-resolution patterns can be printed by a commercial ink-jet printer using pure water as ink and eliminated by dimethyl sulfoxide (DMSO) vapor fuming with high reversibility.

## Results

### Material design and preparation.
The deep-blue OURTP materials were prepared by using cyanuric acid (CA) as a universal host owing to its abundant interaction positions that can efficiently suppress the luminescence quenching effects and non-radiative decay processes, and phthalic acid derivatives as the guest molecules because they can interact effectively with CA for single molecular dispersion and suppressed non-radiative vibration through H-bonding (Fig. 1a)[24,25]. Moreover, CA has very high lowest singlet ($S_1$) and triplet ($T_1$) excited state energies[26] to support efficient energy transfer and function as active host to sensitize the triplet excited state of the guest. CA also has high optical inertness[27] for both photoexcitation and luminescence extraction of guest's emission. Specifically, the unique cyclic amide structure of CA contains three N atoms and three carbonyl groups with lone-pair-electrons to facilitate the spin-forbidden intersystem crossing (ISC) for facile population of $T_1$ state as well as three H-bond donor and three H-bond acceptor positions for constructing H-bonding cross-linked networks with other molecules to efficiently suppress non-radiative decay channels. Trimesic acid (TMA) with three carboxyl moieties which can facilitate the $n–\pi^*$ transition, spin–orbit coupling, and molecular interactions with CA is chosen as the guest molecule (Supplementary Fig. 1). It should be noted that TMA crystal shows OURTP with lifetime of 0.15 s and PhQY of 2.7% at 524 nm, suggesting its intrinsic OURTP nature without the help of other materials[28]. After being doped into CA at a low concentration of 5 wt‰ via simply ultrasonicating aqueous mixture of TMA and CA at room temperature (Fig. 1b), the as-prepared powder of CT5-0 after removing the solvent under vacuum at 40 °C for 24 h exhibits long phosphorescent lifetime of 1.13 s and PhQY of 9.3% under ambient conditions (Fig. 1c). Interestingly, this efficient OURTP is deep blue (406 nm), which is very different to its intrinsic green emission. More interestingly, when 20 wt% water was added into, both phosphorescent lifetime and efficiency of the resulted powder (CT5-20) are significantly enhanced, reaching 1.67 s and 46.1%, respectively (Fig. 1d, e). To the best of our knowledge, both the lifetime and PhQY are among the highest ones of OURTP[29–31], let alone the hardly available deep-blue organic afterglow (Supplementary Table 1, Supplementary Note 1 and Supplementary Figs. 2–4)[26].

### Photophysical properties of CT powders.
To explore the extraordinary highly efficient deep-blue OURTP, the steady-state and time-resolved photophysical properties of CA, TMA, and CT5-0 were investigated[32]. The optical bandgaps of CA and TMA are very large, showing strong absorption bands before 250 nm and fluorescence peaks around 300 nm (Fig. 2a and Supplementary Fig. 5). TMA also shows a decent phosphorescence band around 400 nm even at single molecular state in ethanol, while the steady-state photoluminescence (PL) spectrum of CA is dominated by the blue phosphorescence. Thus, both CA and TMA are capable of populating $T_1$ upon photoexcitation for blue phosphorescence with weak brightness, low quantum efficiencies, and short lifetimes (Supplementary Table 2). But, more facile ISC were observed in CA with much weaker fluorescence in the steady-state PL spectrum (Fig. 2a) and larger spin–orbital coupling (SOC) constants from the Dalton calculations (Supplementary Table 3). Excitingly, when CA and TMA are mixed properly, the composite of CT5-0 exhibits almost identical phosphorescence spectrum to that of TMA (Supplementary Fig. 6) but with significantly increased PhQY and elongated lifetime. When water is implemented, this blue phosphorescence band further enhances and reaches the highest at 20 wt% water content (Fig. 2b, c, and Supplementary Fig. 7). The optimal TMA doping concentration is 5 wt‰ (Fig. 2d, e) and the emission band at 406 nm is nearly pure phosphorescence without the component of short-lived fluorescence (Fig. 2f, g, and Supplementary Fig. 8). In addition, although the afterglow intensity is highly dependent on

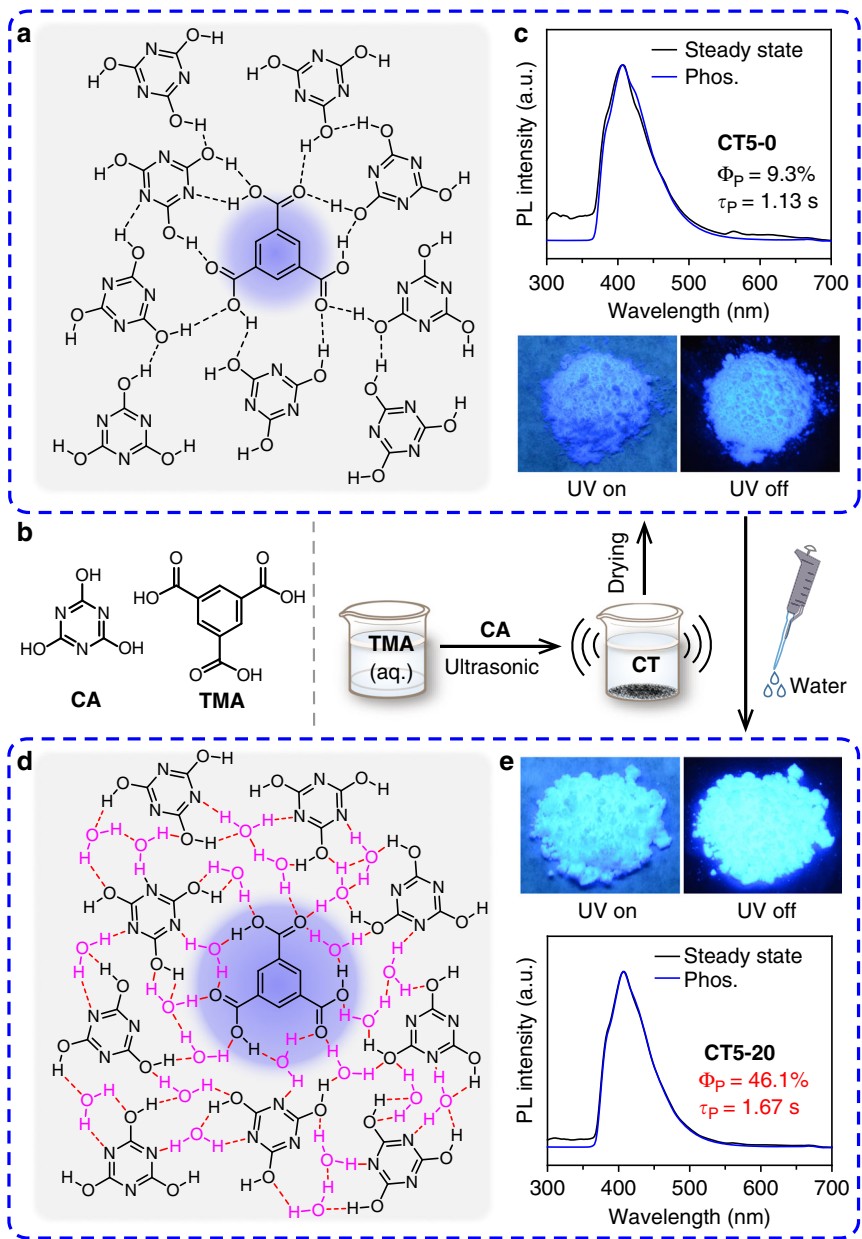

**Fig. 1 Preparation of highly efficient deep-blue OURTP materials. a, d** Molecular interactions between CA and TMA before (**a**) and after (**d**) the water implement. **b** Molecular structures of CA and TMA, and preparing procedures of their composites (CT). **c, e** Steady state and phosphorescent (phos.) spectra and photographs under (UV on) or after (UV off) 254 nm irradiation of CT5-0 (**c**) and CT5-20 (**e**).

water content and doping concentration, the afterglow lifetime is less sensitive to these variations, and nearly invariant under different doping levels of TMA, especially when water content is higher than 20 wt% (Fig. 2h, i, and Supplementary Fig. 9). Therefore, the CT composite reaches the best performance at 5 wt‰ TMA doping and 20 wt% water implement, exhibiting the strongest blue OURTP excitable from 210 to 300 nm with lifetime up to 1.67 s (Fig. 2j and Supplementary Figs. 10, 11).

**Mechanism of the efficient deep-blue OURTP.** To gain further insight into the mechanism of the deep-blue OURTP, temperature-dependent PL measurements were performed. When

temperature drops gradually from 278 to 78 K, the phosphorescence spectrum of CT5-0 remains but the intensity increases constantly along with the elongated lifetime from 1.1 to 2.0 s (Fig. 3a and Supplementary Fig. 12). Similar behavior was observed in CT5-20 and CT5-40 (Fig. 3b and Supplementary Figs. 13, 14) but with limited phosphorescence intensity and lifetime enhancements in these water stimulated samples, suggesting that both low temperature and water implement can suppress the non-radiative decays for longer OURTP lifetime (Supplementary Table 4).

To understand the exact role of water in the composite, differential scanning calorimetry (DSC) measurements, Raman spectra, solid-state nuclear magnetic resonance (NMR) spectra

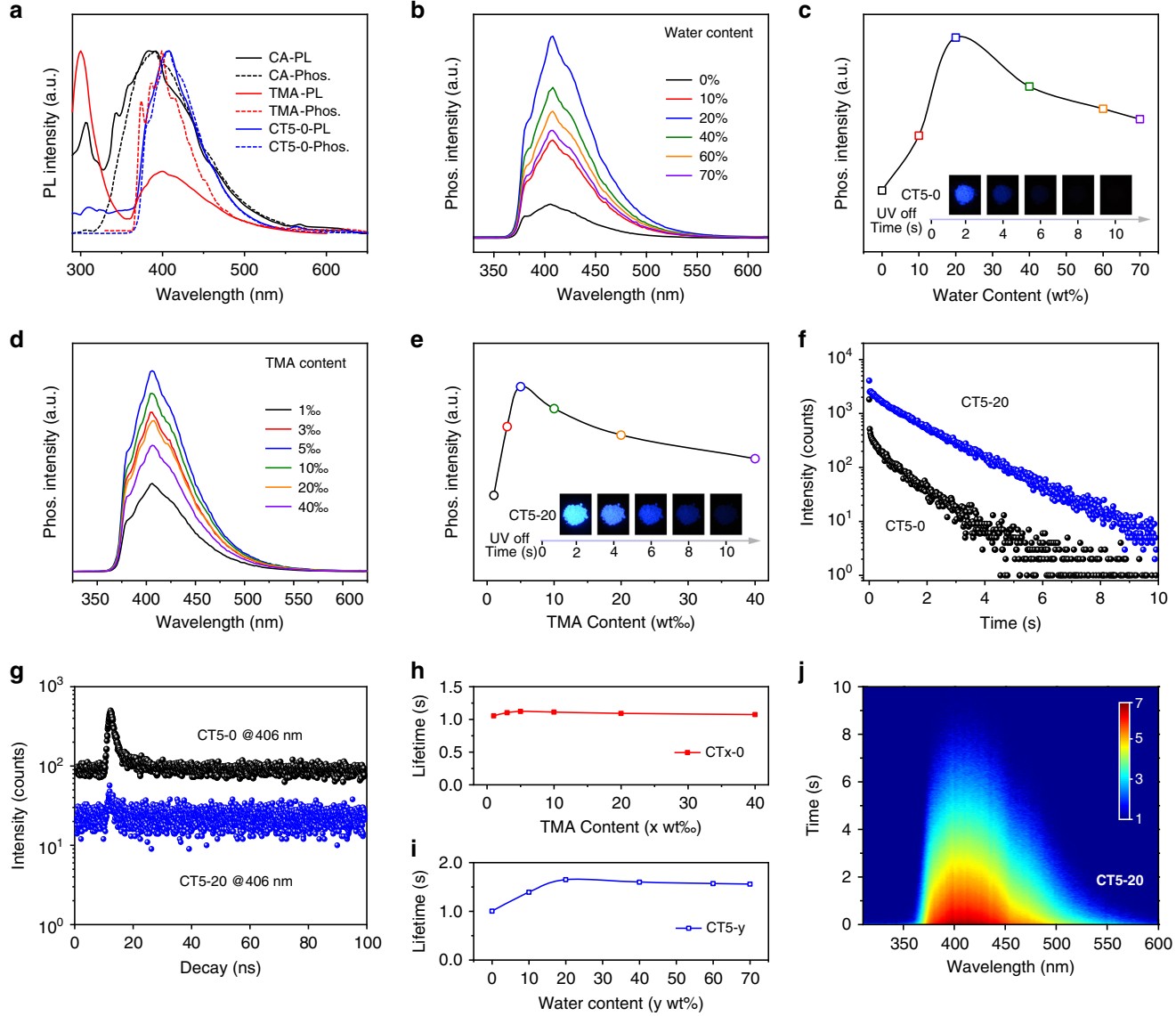

**Fig. 2 Photophysical properties of CT powders at room temperature. a** Steady-state PL (solid line) and phosphorescence (dash line, delay 30 ms) spectra of CT5-0 powder at 298 K as well as TMA in ethanol (10 µM) and CA powder at 77 K. **b–e** Phosphorescent spectra (**b**, **d**) and 406 nm emission intensities (**c**, **e**) of CT5-y with various water contents (y wt%) (**b**, **c**) and dry CTx-0 powder with various doping concentrations (x wt‰) of TMA (**d**, **e**). Insets: photographs of CT5-0 and CT5-20 taken after removing the excitation of 254 nm UV light. **f**, **g** OURTP (**f**) and fluorescence (**g**) decay curves of CT5-0 (black) and CT5-20 (blue) at 406 nm. **h**, **i** OURTP lifetimes of CTx-0 with various TMA concentrations (**h**) and CT5-y with various water contents (**i**). **j** Time-resolved emission scanning spectrum of CT5-20. The excitation wavelength is 248 nm.

and powder X-ray diffraction (XRD) of CT5 with different water contents were investigated. An endothermic melting peak around 7 °C owing to the existence of the freezing bound water and a broad endothermic peak due to the continuous evaporation of water at high temperatures (45–110 °C) were observed in DSC curves when the water content reaches 40% (Fig. 3c)[33]. From the solid-state $^{13}$C-NMR spectra of CT5-0, there is a splitting (152.3 and 149.8 ppm) of the carbon atom of CA at 150.4 ppm (Supplementary Fig. 15 and Supplementary Note 2), indicating the partly H-bonding of the host molecule with or without H-bonding (Fig. 3d)[24]. This splitting disappears after water implement, suggesting that all the polar moieties were strengthened by the H-bonds formed among CA, TMA and water (Fig. 1d). The insertion of water for constructing water-induced H-bonds will lead to the expanding of the composites, which was observed in the XRD curves with reduced 2θ due to the increased interplanar distance of CA after water implement (Fig. 3e).

Further, the H-bond formation was confirmed by the Raman spectra, showing clear shifts of the 1725 and 701 cm$^{-1}$ bands after water addition owing to the intermolecular interactions between the C=O of CA and water molecules, no matter whether doping TMA or not (Fig. 3f). Therefore, an excellent H-bonding network is formed after the introduction the third component of water, resulting in significantly rigidified matrices of the composites with much suppressed non-radiative decays and reduced penetration of oxygen and its quenching effects for greatly improved OURTP performance. Furthermore, H-bonds is beneficial to enhance ISC and Dexter energy transfer[34,35], contributing further to the increased phosphorescence intensity and PhQY after water implement. It should be noted that when other solvents were added to the CT5-0 powder, the phosphorescence intensity and lifetime were generally reduced, confirming that only water can play this bridging role here in forming H-bonds for the matrices rigidification (Supplementary Fig. 16). In addition, the pH can

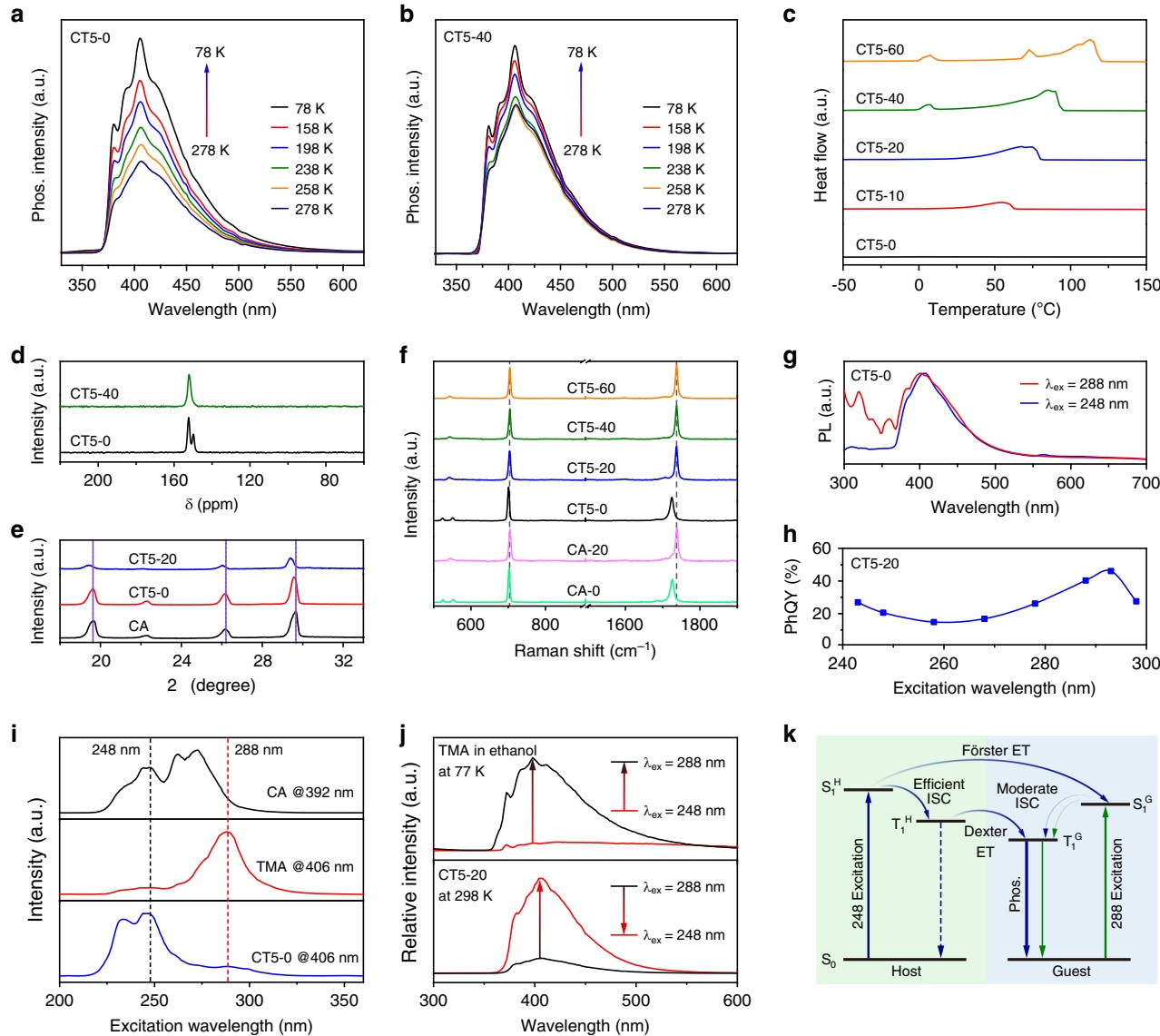

**Fig. 3 Mechanism in realizing the highly efficient deep-blue OURTP. a, b** Temperature-dependent phosphorescence spectra of CT5-0 (**a**) and CT5-40 (**b**) from 278 to 77 K excited at 248 nm. **c** DSC curves of CT5 with various water contents. **d** Solid state $^{13}$C-NMR spectra of CT5-40 and CT5-0. **e** XRD spectra of CA, CT5-0 and CT5-20. **f** Raman spectra of CA with 0 (CA-0) and 20 wt% (CA-20) water and CT5 with various water content. **g** Steady-state PL spectra of CT5-0 excited by 248 (blue) and 288 nm (red) UV-light. **h** PhQY of CT5-20 at different excitation wavelength. **i** Excitation spectra of CA and CT5-0 powders and TMA in ethanol. **j** Phosphorescence spectra of TMA in ethanol at 77 K and CT5-20 at 298 K excited by 248 and 288 nm after 30 ms delay. **k** Proposed mechanism for the highly luminescent deep-blue OURTP.

also influence the OURTP intensity, and the strongest OURTP was observed at the original state when pH = 2 (Supplementary Fig. 17). Either increasing or reducing the pH will lead to lower OURTP, probably due to the difficulties in forming H-bonds at other pH values[27].

Besides the water-sensitive OURTP feature, the emission intensity of the CT composite is also dependent on the excitation wavelength. The almost invisible fluorescence of CT5-0 excited by 248 nm can be apparently observed under excitation of 288 nm (Fig. 3g). And, the PhQY of CT5-20 reaches the highest value of 46.1% at 293 nm excitation with the same emission spectrum (Fig. 3h and Supplementary Fig. 18). From the phosphorescence excitation spectra, CT5-0 has two excitation bands around 248 and 288 nm corresponding to the excitation of CA and TMA, respectively (Fig. 3i). When changing the excitation wavelength from 248 to 288 nm, the phosphorescence intensity of pure TMA rises significantly, but that of CT5-20 decrease dramatically (Fig. 3j). This extraordinary

phenomenon suggests that CA should act as an active host to sensitize TMA by energy transfer (ET) under 248 nm irradiation[36]. It is speculated that 248 nm UV-light excites the host material of CA to $S_1^H$, which quickly transforms to $T_1^H$ to populate $T_1$ of the guest of TMA ($T_1^G$) by Dexter ET, or through Förster ET to form $S_1^G$ and then $T_1^G$ for the long-lived OURTP. The Dexter ET should be the main route during energy transfer from CA to TMA to sensitize $T_1^G$, since the CA is an efficient phosphor with facile ISC for the dominated phosphorescence (Fig. 2a) and the fluorescence of TMA is negligible in the steady-state PL of CT5 composites excited at 248 nm. When CT5 is excited by 288 nm, TMA is directly excited to $S_1^G$, which has moderated ISC to populate $T_1^G$; therefore, both fluorescence and phosphorescence can be observed (Fig. 3g). Since the triplet excitons formation of TMA by CA sensitization under 248 nm excitation are much more efficient than that under 288 nm through direct TMA excitation, the OURTP intensity is much higher than the latter (Fig. 3j). But owing to fewer excited states involved,

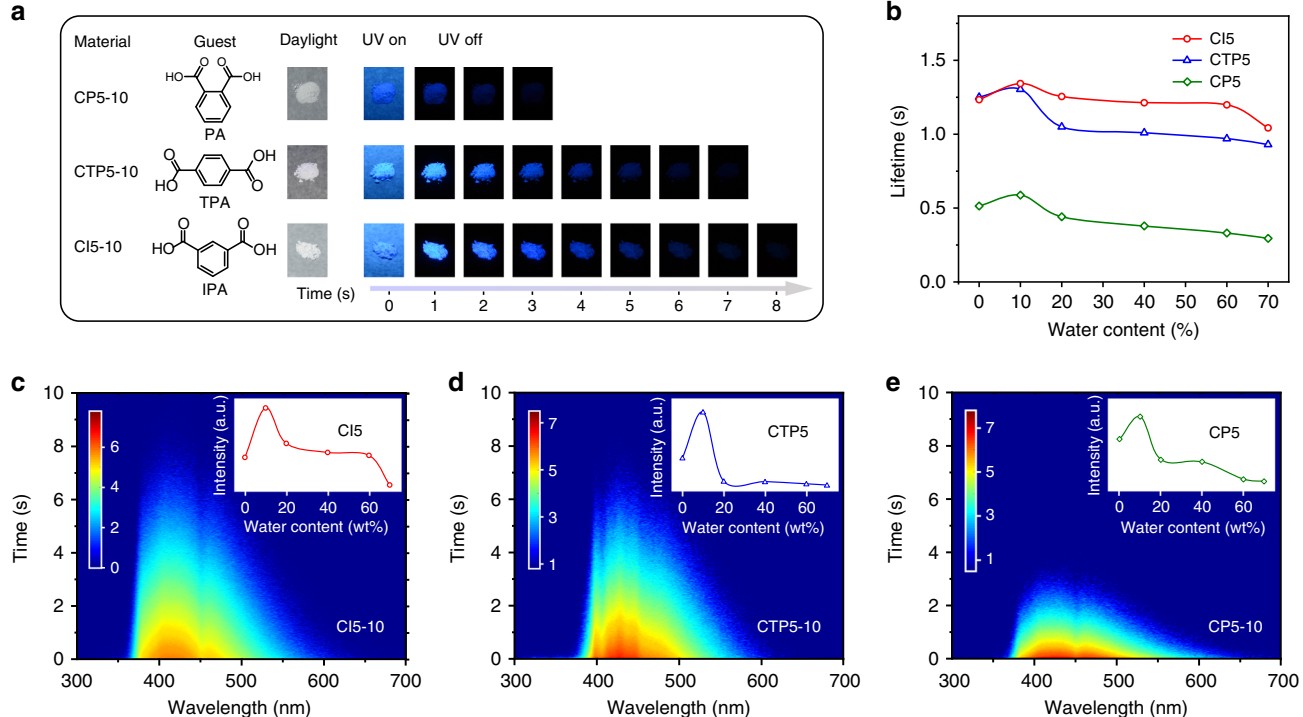

**Fig. 4 Universality of the deep-blue OURTP design strategy. a** Chemical structures of guest molecules and photographs of CI5-10, CTP5-10, and CP5-10 taken under daylight, UV light (UV on) and after removal of excitation (UV off) for several seconds. **b** Lifetimes of CI5, CTP5, and CP5 with various water contents. **c–e** Time-resolved emission scanning spectra of CI5-10 (**c**), CTP5-10 (**d**), and CP5-10 (**e**). Insets: phosphorescence intensities of CI5, CTP5, and CP5 with various water contents.

higher PhQY can be achieved under 288 nm excitation (Fig. 3k). Taking together, the triplet sensitization by the active host and the matrices rigidification effect of water implement should be the two main factors in achieving the highly luminescent and efficient (46.1%) deep-blue OURTP at room temperature.

**Universality of the strategy.** To verify the universality of this strategy using active host and water rigidification for designing high-performance blue/deep-blue organic afterglow materials, we tested other three widely studied benzoic acid derivatives of iso-phthalic acid (IPA), terephthalic acid (TPA), and phthalic acid (PA) as guest molecules (Fig. 4a) to construct the corresponding composites of CI, CTP, and CP (Supplementary Fig. 1 and Supplementary Table 5)[28]. Similarly, the doping in the active host of CA not only results in huge increase in lifetime up to 1.36 s and PhQY of 11.4%, but also blue-shifts the emission bands to 405, 428, and 425 nm for deep-blue OURTP (Supplementary Fig. 19). These OURTP spectra are also identical to the phosphorescent spectra of the guest molecules (Supplementary Fig. 20). Still, water-induced matrices rigidification for enhanced OURTP can be observed in these systems and the best water implement amount is 10 wt%, possibly owing to the fewer H-bonding positions of the guest molecules (Fig. 4b and Supplementary Fig. 21). This also leads to the best OURTP performance of CI5 (Fig. 4c–e) because IPA can form the good H-bonding network with CA, while the *ortho*-substituted PA has difficulties in interacting with host molecules to form H-bonds to rigidify the matrices, resulting in the lowest performance of CP5[37]. Besides, the fewer carboxyl groups of these benzoic acids leads to lower ISC rate and their fluorescence peaks can be observed in the steady-state spectra of these composites (Supplementary Fig. 22).

**Water-jet rewritable encryption paper application.** In light of the extraordinary water responsible OURTP behavior of these

composites, rewritable paper which is crucial for environmental protection and sustainable development can be realized for water-jet printing[38,39]. Specifically, we coat CT5 solution (100 mg·mL⁻¹) in DMSO on a filter paper, followed by removing the solvent to obtain the encryption paper; this water-jet printed patterns can be facilely erased by DMSO vapor fuming at 120 °C for 15 min, representing the successful preparation of rewritable OURTP papers with lifetime-encryption for information storage and anti-counterfeiting (Fig. 5a, Supplementary Fig. 23 and Supplementary Table 6). Other erasing solvents can also work, but DMSO is the most effective (Supplementary Fig. 24 and Supplementary Note 3). Along with the significantly enhanced OURTP after jet-printing using water as ink (Fig. 5b), this rewritable paper has excellent reversibility that can undergo many write/erase cycles (Fig. 5c and Supplementary Table 7). The strong and long-lived phosphorescence of the printed pattern can be conveniently captured by the camera of a mobile phone for the lifetime-resolved encryption of the printed patterns (Fig. 5d). The encrypted afterglow patterns are invisible under daylight and UV light, but can be easily observed after the removal of 254 nm UV irradiation after the repeated water-jet printing and DMSO vapor erasing cycles, presenting an excellent reusable anti-counterfeiting application of complex and high-resolution patterns (Fig. 5e and Supplementary Figs. 25, 26). Moreover, the patterns printed on the encryption paper are highly stable and can last more than 1 month under ambient conditions (Supplementary Fig. 27 and Supplementary Table 8). This deep-blue OURTP can be also used in white organic afterglow lighting with the aid of yellow organic afterglow materials (Supplementary Fig. 28 and Supplementary Note 4).

**Discussion**

In summary, we have established an effective approach to construct a series of heavy-atom-free deep-blue (406–428 nm)

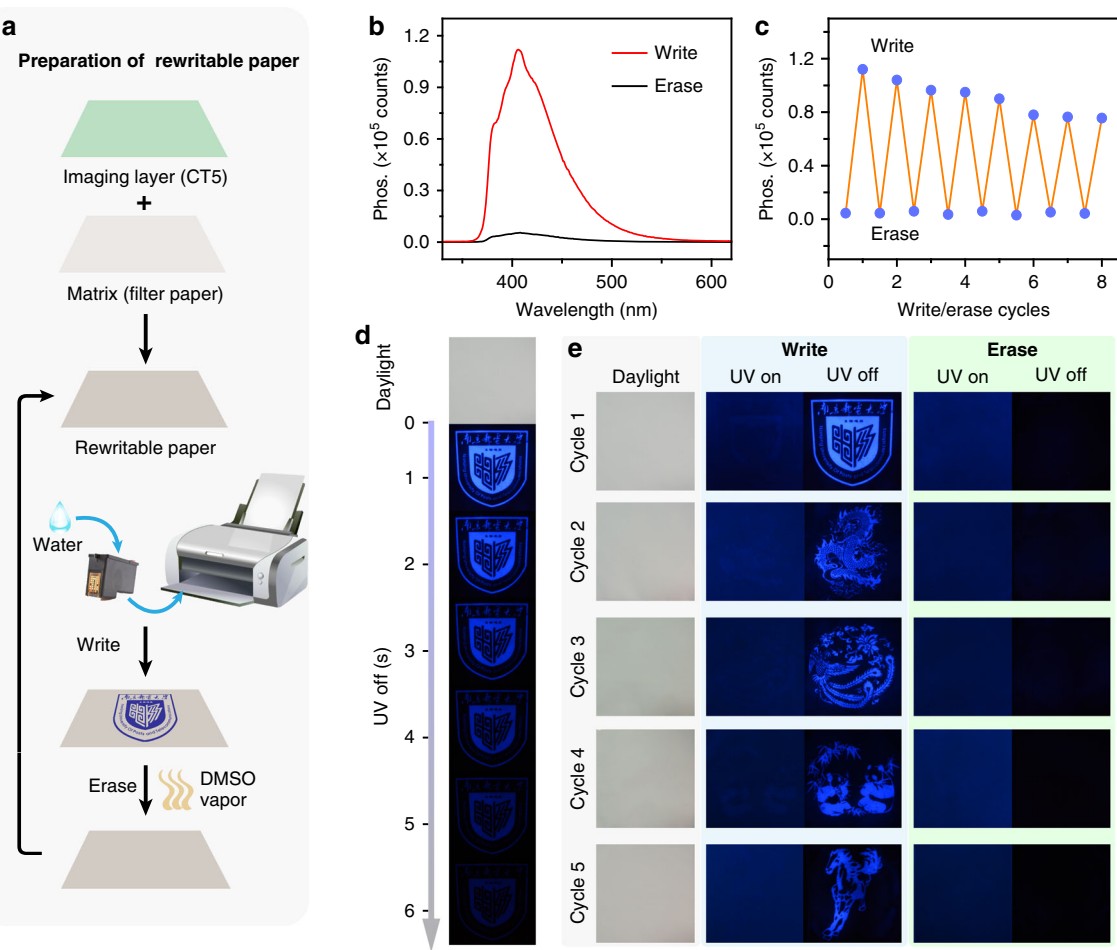

**Fig. 5 Water-jet rewritable encryption papers. a** Preparation of the rewritable encryption paper using pure water as ink. **b** Phosphorescence spectra of rewritable paper after writing and erasing. **c** Reversibility of the rewritable paper. **d** Photographs of the pattern under daylight and after the removal of 254 nm UV light. **e** Photographs taken during five cycles of the write/erase processes under daylight, UV light (UV-on), and after removal of the excitation (UV-off).

organic afterglow materials with PhQYs and lifetimes up to 46.1% and 1.67 s, respectively. The principle of this approach, which involves using active host to sensitize the triplet exciton population via Dexter energy transfer for strong phosphorescence and water implement to rigidify the matrices through H-bonding networks to suppress the non-radiative decays for high-efficient emission, provides a universal and convenient access to design highly luminescent and efficient OURTP observable by naked eyes under ambient conditions. Moreover, with the unique water-responsible OURTP mechanism, we develop a rewritable lifetime-encryption paper using CT5 as substrate and water as ink for the high resolution water-jet printing of any patterns that can last for 1 month and be erased by DMSO vapor conveniently in 15 min. Although there is still much to be learned regarding the exact mechanism in triplet excited state sensitization and stabilization, this host sensitization/water rigidification strategy would hold substantial promise for advancing the development of high-performance blue/deep-blue OURTP materials with high efficiency, strong brightness and long lifetime simultaneously and may inspire future innovative applications with the stimuli-responsive phosphorescent features.

## Methods

**Preparation of CA-based composites**. CA-based deep-blue OURTP composites including CT, CI, CTP, and CP were prepared in a standard procedure. Take CT5 as a typical example. First, 5 mg TMA was dissolved in 5 mL deionized water to obtain a 1.0 mg/mL aqueous solution of TMA. Second, to the 1 mL TMA solution was added 0.2 g CA, and the mixture was ultrasonicated at room temperature for 10 min. Finally, the mixture was dried by removing the solvent of water under vacuum at 40 °C for 24 h to get the target composite of CT5-0 with TMA weight content of 5‰. To obtain CT5-20 with 20 wt% water, 25 μL water was added to the 0.1 g CT5-0 followed by ultrasonication at room temperature for 10 min.

**Fabrication of the rewritable lifetime-encryption paper**. 500 mg CA and 5 mg TMA were dissolved in 5 mL DMSO to afford the coating solution. This solution was uniformly coated on a filter paper and after drying at 100 °C for 2 h, the rewritable lifetime-encryption paper was obtained.

**Water-jet printing of the rewritable paper**. The desired pattern was printed by a commercially available printer (HP DeskJet 1111) driven by a computer using pure water as ink.

**Erase of the rewritable paper**. The pattern printed on the rewritable paper was erased using DMSO vapor generated by heating DMSO to 120 °C. The complete erase of the pattern costs about 15 min.

## Data availability

The data that support the findings of this study are available from the corresponding author upon reasonable request.

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

## Acknowledgements

This work is supported by the National Natural Science Foundation of China (21772095, 91833306, 21674049, 61875090, 91733302, and 61935017), Key giant project of Jiangsu Educational Committee (19KJA180005), the fifth 333 project of Jiangsu Province of China (BRA2019080), China Postdoctoral Science Foundation (2020M671460), Jiangsu Planned Projects for Postdoctoral Research Funds (2020Z137), 1311 Talents Program of Nanjing University of Posts and Telecommunications (Dingshan) and the Science and Technology Development Fund Macao SAR (File no. FDCT-0044/2020/A1).

## Author contributions

S.X., W.W., and R.C. conceived the experiments. S.X., C.Z., W.H. G.X., and R.C. wrote the manuscript. S.X., W.W., and J.Z. were primarily responsible for the material preparation. W.W., J.Z., H.L., and S.W. measured the photophysical properties and structure characterizations. S.W. performed the theoretical calculations. C.S. measured the Raman spectra. All authors contributed to the discussion of the results.

## Competing interests

The authors declare no competing interests.
