## [Peer Review File · Nature Communications]

Reviewers' Comments:

Reviewer #1:

Remarks to the Author:

In this manuscript, the authors reported a strategy to achieve efficient blue organic afterglow by designing the host-guest systems using different active hosts (e.g., trimesic acid (TMA), isophthalic acid (IPA), terephthalic acid (TPA) and phthalic acid (PA)) for triplet state sensitization, and cyanuric acid (CA) as the guest for matrices rigidification. As a result, a series of heavy-atom-free deep-blue organic afterglow composite materials were obtained and exhibited superior deep-blue room temperature phosphorescence (RTP) (phosphorescence quantum efficiency (PQYs) to be 46.1% and lifetime up to 1.67 s). Further study indicated that the active hosts to sensitize the triplet exciton population via Dexter energy transfer and the water implement to rigidify the matrices through H-bonding networks to suppress the non-radiative decays are responsible for the strong and efficient RTP emission. Finally, applications of the prepared material in information encryption was demonstrated. This study is interesting and may be publishable in Nature Communication after some revisions.

1. The authors confirmed the formation of H-bond among CA, TMA and water by the solid-state ¹³C-NMR and Raman spectra (Page 4). However, the description in the manuscript does not fully clarify this conclusion, please provide more detailed discussion and explanation.
2. The authors claimed that "Since the triplet excitons formation of TMA by CA sensitization under 248 nm excitation are much more efficient than that under 288 nm through direct TMA excitation" in page 5. It's not clear to me how did the authors draw such conclusion.
3. Does the pH of water affect the RTP performance?
4. As a key data, the phosphorescence quantum yield (PQY) is critical for afterglow materials. However, I doubted the accuracy of the reported PQY, because it is not possible to separate RTP from the whole emission (which may include fluorescence as well). Therefore, the detailed measurement procedure and reasonableness for PQY should be provided.
5. The authors emphasized that blue luminescent (including fluorescent and phosphorescent) materials are indispensable for solid-state lighting and full-color display technologies in organic optoelectronics. However, they failed to verify the importance of their materials in such fields. I am wondering how to use the as-prepared materials in these fields and how much value they could have.

Reviewer #2:

Remarks to the Author:

In this manuscript, the authors present an efficient strategy to design deep-blue organic afterglow materials with very long luminescence lifetime. Interestingly, both lifetime and quantum yield (PQY) of the afterglow can be enhanced simultaneously by water implement, which is contrary to the general effects of water on photo-luminescence. The organic afterglow performance is impressive with PQY up to 46.1% and a water-jet printing for rewritable anti-counterfeiting paper was developed, illustrating a new approach for high-tech applications of organic afterglow materials. Thus, I suggest the acceptance of this manuscript after minor revision upon addressing the following concerns.

1. In mechanism section, the authors found that Dexter energy transfer is the main sensitization route through energy transfer from CA to TMA. How to support the effective Dexter energy transfer, since the doping concentration of TMA is low to 5 wt%?
2. The authors claim that CA has efficient ISC process while TMA has moderate ISC. Any solid evidence can prove this?
3. The boiling point of DMSO used for erase of rewritable paper is too high for the practical use. Is there any solution can replace DMSO with lower boiling point?
4. The organic afterglow and deep blue photoemission are recent hot topics in materials and chemical science, to arouse a broad interest from readers in this field, several strongly related works can be added as references, such as molecular afterglow materials with ultralong emission

lifetimes (Angew. Chem. Int. Ed. 2019, 58, 15128; Mater. Horiz. 2014, 1, 46). These literatures may be helpful to the readers for better understand the development of organic phosphorescent materials and their photofunctional applications.

5. Some expressions should be carefully considered, e.g. 'ink-free rewritable OURTP paper' and 'using water as ink' in the application section.

Reviewer #3:

Remarks to the Author:

In this paper, the authors reported a design strategy for deep-blue afterglow materials with long lifetimes and high phosphorescence quantum yields through guest sensitization and matrices rigidification using active host and water. As authors mentioned, blue room-temperature phosphorescence from organic molecules is quite interesting, but never reported. Present mechanisms include stabilized triplet states of H aggregation or charge trapping and releasing of triplet exciton formation. These two mechanism can be hardly used to form blue phosphors, since aggregation and charge transfer can induce emission red shifts. In this paper, the authors proposed a new mechanism, doping organic phosphorescence emitters in rigid matrixes. I think it is very important that since the phosphorescence is from single molecule, it is feasible for material design with an accurate purpose for color tuning rather. The authors also proposed a rewritable paper that can be printed using water as ink to print any anti-counterfeiting patterns, even high-resolution pictures. The performance reported in this paper is very high, e.g. quantum yield of 46.1% and lifetime of 1.67 s. So, I suggest this manuscript would be acceptable after a minor revision.

1. As shown in Figs.1c, 1e and 2b, the phosphorescence intensity and ratio increase significantly after water addition. The authors attributed this phenomenon to the suppression of non-radiative decays. Any other reasons can be also considered?

2. Recently, the device applications of purely organic phosphorescence materials have attracted rising attentions owing to the theoretically 100% IQE. Can CT5 used as emitters for electroluminescence as it has high PQY of 46%?

3. Can other solvents like chlorobenzene (generally used in OLED) used for erasing the anti-counterfeiting patterns? How to choose the erase solvent?

4. After addition of water, the distance between CA and TMA would be elongated. However, it is showed that the energy transfer between them is not influenced. So, hydrogen bond networks would be also involved in the Dexter energy transfer. Actually, recent investigation showed that the hydrogen bond networks can be used to control the processes of carrier transport and exciton formation (Adv. Funct. Mater. 2020, 30, 1908568). The authors are suggested to add some brief explanations in this aspect.

5. Some errors or writing styles must be corrected, e.g. legend of Fig. 3e is not consistent with the figure. The abbreviation of "PQY" for phosphorescence quantum yield is quite confused. I would like to suggest "PhQY".

Re: Design of Highly Efficient Deep-blue Organic Afterglow through Guest Sensitization and Matrices Rigidification (Manuscript ID: NCOMMS-20-11975).

Thank you so much for your letter dated 20-May-2020 regarding our manuscript entitled: **“Design of Highly Efficient Deep-blue Organic Afterglow through Guest Sensitization and Matrices Rigidification”** (Authors: *Shen Xu, Wu Wang, Hui Li, Jingyu Zhang, Runfeng Chen, Shuang Wang, Chao Zheng, Guichuan Xing, Chunyuan Song, Wei Huang*). The current submission is a revised one according to the editor/reviewer’s comments.

We have carefully gone through the comments of you and all the reviewers and made necessary changes to the manuscript. All these changes have been highlighted in red in the review-only version of the revised manuscript. Our responses to the comments of the reviewers are as follows.

The reply is indicated by black letters, while the comments by the reviewers are indicated by blue italic letters. We have documented the changes we have made to the original manuscript.

Reviewer comments:**Reviewer: #1**

In this manuscript, the authors reported a strategy to achieve efficient blue organic afterglow by designing the host-guest systems using different active hosts (e.g., trimesic acid (TMA), isophthalic acid (IPA), terephthalic acid (TPA) and phthalic acid (PA)) for triplet state sensitization, and cyanuric acid (CA) as the guest for matrices rigidification. As a result, a series of heavy-atom-free deep-blue organic afterglow composite materials were obtained and exhibited superior deep-blue room temperature phosphorescence (RTP) (phosphorescence quantum efficiency (PQYs) to be 46.1% and lifetime up to 1.67 s). Further study indicated that the active hosts to sensitize the triplet exciton population via Dexter energy transfer and the water implement to rigidify the matrices through H-bonding networks to suppress the non-radiative decays are responsible for the strong and efficient RTP emission. Finally, applications of the prepared material in information encryption was demonstrated. This study is interesting and may be publishable in Nature Communications after some revisions.

Response: Thanks for professional comments and kind recommendation of our work. We have revised the manuscript accordingly.

Comments 1. *The authors confirmed the formation of H-bond among CA, TMA and water by the solid-state ¹³C-NMR and Raman spectra (Page 4). However, the description in the manuscript does not fully clarify this conclusion, please provide more detailed discussion and explanation.*

Response: Thanks for the kind reminding. The description of the solid state ¹³C-NMR and Raman spectra to confirm the formation of H-bond has been revised to fully clarify the related conclusion in the main text and supporting information. Detailed discussion and explanation are as follows. In the main text, “From the solid-state ¹³C-NMR spectra of CT5-0, there is a splitting (152.3 and 149.8 ppm) of the carbon atom of CA at 150.4 ppm (Supplementary Scheme 5), indicating the partly H-bonding of the host molecule with or without H-bonds (Fig. 3d).” “The H-bond formation was confirmed further by the Raman spectra, showing clear shifts of the 1725 and 701 cm⁻¹ bands after water addition owing to the intermolecular interactions between the C=O of CA and water molecules, no matter whether doping TMA or not (Fig. 3f).” In the

Supplementary Information, “**CA** with 3 equivalent carbon atoms has only a single peak at around 150 ppm in ^{13}C -NMR (Supplementary Scheme 5). However, in solid-state ^{13}C -NMR spectrum of **CT5-o**, two peaks at 152.3 and 149.8 ppm can be observed, corresponding to carbon atoms with and without H-bonding in **CA**. After water implement, only a single peak at 152.1 ppm appears, indicating that all carbon atoms are in H-bonds.” Thanks a lot.

Supplementary Scheme. 5. ^{13}C -NMR spectrum of **CA** in DMSO-d_6 .

Comments 2. *The authors claimed that “Since the triplet excitons formation of **TMA** by **CA** sensitization under 248 nm excitation are much more efficient than that under 288 nm through direct **TMA** excitation” in page 5. It’s not clear to me how did the authors draw such conclusion.*

Response: Thanks for the kind reminding. In Fig.3i, two excitation peaks of **CT5-o** located at 248 and 288 nm were observed corresponding to the excitation of **CA** and **TMA**, respectively. When **CT5-o** is excited by 248 nm, **CA** is excited and most excitons in **CA** transfer to triplet state via intersystem crossing first, then to **TMA** through Dexter energy transfer, resulting in nearly invisible fluorescent emission band around 300 nm (blue line in Fig. 3g). Under 288 nm excitation, **TMA** is directly excited, leading to the more apparent fluorescent emission band before 350 nm (red line in Fig. 3g) owing to its relatively weaker intersystem crossing. Moreover, as shown in Fig. 3j, the phosphorescent intensity of **CT5-20** excited by 248 nm is much higher than that excited by 288 nm. Consequently, we conclude that “triplet excitons formation of **TMA** by **CA** sensitization under 248 nm excitation are much more efficient than that under 288 nm through direct **TMA** excitation”. We have revised the corresponding discussions to make it

clearer. Thanks again.

Fig. 3 g, Steady-state PL spectra of CT5-o excited by 248 (blue) and 288 nm (red) UV-light. h, Excitation spectra of CA and CT5-o powders and TMA in ethanol. i, Phosphorescence spectra of TMA in ethanol at 77 K (top) and CT5-20 at 298 K (bottom) excited by 248 and 288 nm after 30 ms delay.

Comments 3. Does the pH of water affect the RTP performance?

Response: Thanks for the kind reminding and good suggestion. In the revised manuscript, we investigated the effects of pH on the RTP performance by adding HCl or NaOH into the TMA solution to control the pH (Supplementary Figure 10). It was found that RTP performance is dependent on the pH. The phosphorescence intensity reaches the maximum at pH = 2, and it is fully quenched when pH > 5. We have updated the related discussions in the revised manuscript.

Thanks again.

Supplementary Figure 10. Phosphorescent spectra of **CT5-20** prepared by adding **CA** to the aqueous solution of **TMA** with different pH values under ambient conditions. A delay time of 30 ms was applied. Inset: phosphorescence intensities at 406 nm with different pH values of **TMA** solution.

Comments 4. As a key data, the phosphorescence quantum yield (PQY) is critical for afterglow materials. However, I doubted the accuracy of the reported PQY, because it is not possible to separate RTP from the whole emission (which may include fluorescence as well). Therefore, the detailed measurement procedure and reasonableness for PQY should be provided.

Response: Thanks for the professional comment. Fluorescence of both **CA** and **TMA** are located before 350 nm which is in line with the references (*Nat. Photonics* **2019**, *13*, 406-411; *Adv. Opt. Mater.* **2016**, *4*, 897-905). Thus, the fluorescence can be easily distinguished from the phosphorescence of **CT5-20** peaked at 406 nm. Besides, the fluorescence of **CT5-20** is very weak and almost indistinguishable compared to the phosphorescence in its steady-state PL spectrum (Fig. 1e) and fluorescence decay curves of **CT5-20** (inset of Fig. 2d). Therefore, we can accurately measure the PQY by only integrating the emission band in the range from 365 to 620 nm which are only the phosphorescent emission without need to separate RTP from the whole emission. We have clarified this point in the revised Supplementary Information. Many thanks.

Comments 5. The authors emphasized that blue luminescent (including fluorescent and phosphorescent) materials are indispensable for solid-state lighting and full-color display technologies in organic optoelectronics. However, they failed to verify the importance of their

materials in such fields. I am wondering how to use the as-prepared materials in these fields and how much value they could have.

Response: We totally agree with the concern of this reviewer. We think that the lifetime-encrypted anti-counterfeiting technique used in rewritable paper should be a kind of display application. For the solid-state lighting, we doped 1 wt% **DPhCzT** (a yellow afterglow material reported previously, *Nat. Mater.*, **2015**, 14, 685-690) in **CT5-20**. Such obtained composite material exhibits white afterglow after the excitation of 254 nm UV light, indicating the potential application in white lighting with the assistance of a UV light emitting diode lamp (Supplementary Figure 21). Thanks a lot.

Supplementary Figure 21. Phosphorescent spectrum of 1 wt% **DPhCzT**-doped **CT5-20** collected with a delay time of 30 ms after the excitation of the 254 nm UV light. Inset: photograph of the composite after removing the 254 nm excitation source.

Reviewer: #2

In this manuscript, the authors present an efficient strategy to design deep-blue organic afterglow materials with very long luminescence lifetime. Interestingly, both lifetime and quantum yield (PQY) of the afterglow can be enhanced simultaneously by water implement, which is contrary to the general effects of water on photo-luminescence. The organic afterglow performance is impressive with PQY up to 46.1% and a water-jet printing for rewritable anti-counterfeiting paper was developed, illustrating a new approach for high-tech applications of organic afterglow materials. Thus, I suggest the acceptance of this manuscript after minor revision upon addressing the following concerns.

Response: Thanks for the kind recommendation of our work. We have carefully revised the manuscript.

Comments 1. *In mechanism section, the authors found that Dexter energy transfer is the main sensitization route through energy transfer from CA to TMA. How to support the effective Dexter energy transfer, since the doping concentration of TMA is low to 5 wt%?*

Response: Thanks for the professional comment. Indeed, reducing the doping concentration can effectively suppress the Dexter energy transfer owing to the short lifetime (several microseconds) of the triplet excitons of hosts as found in organic light-emitting diodes. Nevertheless, phosphorescent lifetime of CA is much longer and is up to 0.36 s, which can support the much longer diffusion length of the triplet excitons of CA to sensitize the guest molecules via Dexter energy transfer. Moreover, it has been reported that H-bonds can act as the good transfer media of triplet excitons (*Science* **1995**, 269, 1409-1413; *J. Am Chem Soc* **1995**, 117, 704-714; *J. Phys Chem A* **2008**, 112, 3865-3869). Therefore, the long phosphorescent lifetime of CA and abundant H-bond networks support the efficient Dexter energy transfer from CA to TMA in a low doping concentration. We have updated related discussions in the revised manuscript. Thanks again.

Comments 2. The authors claim that **CA** has efficient ISC process while **TMA** has moderate ISC. Any solid evidence can prove this?

Response: Thanks for the professional question. As shown in Fig.2a in the main text, the PL spectrum of **TMA** is composed mainly by fluorescence emission band with a phosphorescence tail, while the phosphorescence of **CA** dominates its PL spectrum, indicating that **CA** can form triplet excitons much easier than **TMA**. Further, we calculated the spin-orbital coupling (SOC) constants of **CA** and **TMA**. As shown in Supplementary Table 3, SOC constants of **CA** are much larger than that of **TMA**, which confirms the photophysical observations. Therefore, we can conclude that **CA** has efficient ISC process while **TMA** has moderate ISC. These new data have been involved in the revised manuscript. Many thanks.

Fig. 2 a, Steady-state PL (solid line) and phosphorescence (dash line, delay 30 ms) spectra of **CT5-o** powder at 298 K as well as **TMA** in ethanol (10 μ M) and **CA** powder at 77 K.

Supplementary Table 3. Singlet-triplet splitting energy ($E_{S_1}-E_{T_n}$) and SOC constant from S_1 to T_n of **TMA** and **CA**. The efficient intersystem crossing channels with $E_{S_1}-E_{T_n} < 0.37$ eV were highlighted in red.

Molecule	Transition	$E_{S_1}-E_{T_n}$ (eV)	SOC (cm^{-1})
TMA	$S_1 \rightarrow T_1$	0.95	10.67
	$S_1 \rightarrow T_2$	0.41	5.30
	$S_1 \rightarrow T_3$	0.38	2.35
	$S_1 \rightarrow T_4$	0.31	2.20
	$S_1 \rightarrow T_5$	0.24	2.18
	$S_1 \rightarrow T_6$	0.23	4.45
	$S_1 \rightarrow T_7$	-0.02	2.41
	$S_1 \rightarrow T_8$	-0.60	7.79
	$S_1 \rightarrow T_9$	-0.61	9.64
	$S_1 \rightarrow T_{10}$	-0.73	6.28
CA	$S_1 \rightarrow T_1$	1.04	26.42
	$S_1 \rightarrow T_2$	0.47	13.61
	$S_1 \rightarrow T_3$	0.47	9.51
	$S_1 \rightarrow T_4$	0.31	15.35
	$S_1 \rightarrow T_5$	0.26	2.98
	$S_1 \rightarrow T_6$	0.26	10.60
	$S_1 \rightarrow T_7$	0.10	0.59
	$S_1 \rightarrow T_8$	0.06	4.06
	$S_1 \rightarrow T_9$	-1.23	1.04
	$S_1 \rightarrow T_{10}$	-1.23	6.48

Comments 3. The boiling point of DMSO used for erase of rewritable paper is too high for the practical use. Is there any solution can replace DMSO with lower boiling point?

Response: Thanks for the good suggestion. We have tested other solvents with relatively lower boiling points. As shown in Supplementary Figure 17, these solvents show poorer erasing performance. It is speculated that introduction of DMSO can damage the H-bond network constructed by water molecules and partly dissolve **TMA** molecules to quench the phosphorescence. Therefore, DMSO can fully erase the phosphorescence in a short time. In addition, DMSO just needs to be heated to 120°C to erase the printed patterns, which is much lower than its boiling point. Thanks again. We have involved these data and discussions in the revised manuscript.

Supplementary Figure 17. Phosphorescent spectra of rewritable paper at initial state, after water writing, and erased by toluene (a), chlorobenzene (b) and DMF (c) vapor fuming for different time. The excitation wavelength is 248 nm and the delay time is 30 ms.

Comments 4. *The organic afterglow and deep blue photoemission are recent hot topics in materials and chemical science, to arise a broad interest from readers in this field, several strongly related works can be added as references, such as molecular afterglow materials with ultralong emission lifetimes (Angew. Chem. Int. Ed. 2019, 58, 15128; Mater. Horiz. 2014, 1, 46). These literatures*

may be helpful to the readers for better understand the development of organic phosphorescent materials and their photofunctional applications.

Response: Thanks for the important information. We have cited these literatures in the revised manuscript.

Comments 5. *Some expressions should be carefully considered, e.g. 'ink-free rewritable OURTP paper' and 'using water as ink' in the application section.*

Response: We are sorry for the improper expressions. We have carefully gone through the whole manuscript and corrected these expressions. Thanks for the kind reminding.

Reviewer: #3

In this paper, the authors reported a design strategy for deep-blue afterglow materials with long lifetimes and high phosphorescence quantum yields through guest sensitization and matrices rigidification using active host and water. As authors mentioned, blue room-temperature phosphorescence from organic molecules is quite interesting, but never reported. Present mechanisms include stabilized triplet states of H aggregation or charge trapping and releasing of triplet exciton formation. These two mechanism can be hardly used to form blue phosphors, since aggregation and charge transfer can induce emission red shifts. In this paper, the authors proposed a new mechanism, doping organic phosphorescence emitters in rigid matrixes. I think it is very important that since the phosphorescence is from single molecule, it is feasible for material design with an accurate purpose for color tuning rather. The authors also proposed a rewritable paper that can be printed using water as ink to print any anti-counterfeiting patterns, even high-resolution pictures. The performance reported in this paper is very high, e.g. quantum yield of 46.1% and lifetime of 1.67 s. So, I suggest this manuscript would be acceptable after a minor revision.

Response: We appreciate the reviewer's professional and thoughtful comments and kind recommendation of our work.

Comments 1. *As shown in Figs.1c, 1e and 2b, the phosphorescence intensity and ratio increase significantly after water addition. The authors attributed this phenomenon to the suppression of non-radiative decays. Any other reasons can be also considered?*

Response: Thanks for kind reminding. Suppression of non-radiative decays is an important reason for the enhancement of phosphorescence (reduced k_{nr}). Besides that, H-bond-induced matrices rigidification is another one. The water implement for the strengthened H-bond network reduces the exposing area of host and guest molecules, thus preventing triplet quenching effect caused by environmental oxygen (reduced k_q). Moreover, the formation of H-bonds can facilitate intersystem crossing and reduce the triplet quenching for more efficient OURTP (*ACS Appl. Mater. Interfaces* **2020**, *12*, 20765), thus enhancing the emission of phosphorescence. We have updated the related discussions in the revised manuscript. Thanks a lot.

Comments 2. *Recently, the device applications of purely organic phosphorescence materials have attracted rising attentions owing to the theoretically 100% IQE. Can CT5 used as emitters for electroluminescence as it has high PQY of 46%?*

Response: Thanks for the good suggestion. We have tried to use CT5 as emitting layer in organic light emitting diode (OLED) devices, but we cannot realized the electroluminescent (EL) afterglow. In the devices with the structure configuration of ITO/PEDOT:PSS/TAPC/Ir(ppz)₃/CT5/TSPO1/LiF/Al, PEDOT:PSS and TAPC act as hole injection and transporting layers, while Ir(ppz)₃ and TSPO1 serve as exciton blocking layer and electron transporting layer, respectively. The electroluminescent (EL) spectrum of this device is in line with the EL spectrum of TAPC, indicating improper recombination zone or exciton transfer from the emitting layer to the hole-transporting layer. Besides, the device performance drops dramatically when increasing the thickness of the emitting layer, demonstrating the poor carrier-transporting ability of the host molecule of CA in CT5. We further fabricated a single-layer device using the following structure: ITO/MoO_x (3 nm)/CT5 (50 nm)/LiF (0.7 nm)/Al.

Unfortunately, this device fails to be turned on. We think there are several factors may be ascribed to the failure: 1) it is hard to choose hole/electron transporting materials with high enough triplet energy levels to prevent exciton back transfer; 2) the water implement in OLED device is impossible, leading to low PQY (9.3% for **CT5-0**); 3) unsatisfied charge carrier transporting ability of **CT5** results in very high driving voltage. We should solve these issue in the future to realize the organic afterglow OLEDs. Many thanks.

Fig. 1 for review only. OLED performance using **CT5** as emitter. **a**, current density-luminance-voltage (*J-V-L*) curves (inset: electroluminescent spectrum). **b**, current efficiency (CE) and power efficiency (PE) curves (inset: device structure and energy diagram).

Comments 3. Can other solvents like chlorobenzene (generally used in OLED) used for erasing the anti-counterfeiting patterns? How to choose the erase solvent?

Response: Thanks for the thoughtful suggestion. We tested other solvents like N,N-dimethylformamide (DMF), chlorobenzene and toluene, but the afterglow cannot be fully erased after 120 min (Supplementary Figure 17). In contrast, DMSO can fully erase the phosphorescence in a short time at a much lower temperature of 120°C than its boiling point. To choose the erase solvent, there are two aspects should be considered: 1) ability to break the H-bond network and 2) solubility of **TMA** molecules to quench the phosphorescence.

Supplementary Figure 17. Phosphorescent spectra of rewritable paper at initial state, after water writing, and erased by toluene (a), chlorobenzene (b) and DMF (c) vapor fuming for different time. The excitation wavelength is 248 nm and the delay time is 30 ms.

Comments 4. After addition of water, the distance between CA and TMA would be elongated. However, it is showed that the energy transfer between them is not influenced. So, hydrogen bond networks would be also involved in the Dexter energy transfer. Actually, recent investigation showed that the hydrogen bond networks can be used to control the processes of carrier transport

and exciton formation (Adv. Funct. Mater. 2020, 30, 1908568). The authors are suggested to add some brief explanations in this aspect.

Response: We appreciate the professional and thoughtful suggestion. We totally agree that H-bond networks can enhance the Dexter energy transfer. The multiple effects of the H-bond networks have been explained in the revised manuscript. Many thanks.

Comments 5. Some errors or writing styles must be corrected, e.g. legend of Fig. 3e is not consistent with the figure. The abbreviation of “PQY” for phosphorescence quantum yield is quite confused. I would like to suggest “PhQY”.

Response: We are sorry for these mistakes. We have checked the whole manuscript carefully and revised these errors. We also thank the kind suggestion and changed the abbreviation of phosphorescence quantum efficiency.

If there are any further queries, please feel free to contact me. I can be reached by E-mail: iamrfchen@njupt.edu.cn or by Fax: (+86) 25 8586 6396.

Yours sincerely,
Run-Feng CHEN
Key Laboratory for Organic Electronics & Information Displays (KLOEID)
Nanjing University of Posts & Telecommunications
9 Wenyuan Road, Nanjing 210023, China
Tel/Fax: +86 25 8586 6396
Cell: +86 15366190470
E-mail: iamrfchen@njupt.edu.cn

Reviewers' Comments:

Reviewer #1:

Remarks to the Author:

The authors had responded my concerns properly in this revised manuscript, so I can recommend its acceptance and publication.

Reviewer #2:

Remarks to the Author:

In my view, the authors have answered all the questions from reviewers and revised related points, and thus this revised work can be published as it is.

Reviewer #3:

Remarks to the Author:

All my concerns were well-addressed in the revised manuscript. The mechanism of blue RTP was clarified. The performance of the materials is quite impressive. These results are interesting and would be referable for physical science community. So, this paper was improved to meet the high standard of Nat Commun, and can be published as it.

Reviewer comments:

Reviewer: #1

The authors had responded my concerns properly in this revised manuscript, so I can recommend its acceptance and publication.

Response: Many thanks for the kind recommendation!

Reviewer: #2

In my view, the authors have answered all the questions from reviewers and revised related points, and thus this revised work can be published as it is.

Response: We are grateful for the recommendation!

Reviewer: #3

All my concerns were well-addressed in the revised manuscript. The mechanism of blue RTP was clarified. The performance of the materials is quite impressive. These results are interesting and would be referable for physical science community. So, this paper was improved to meet the high standard of Nat Commun, and can be published as it.

Response: We appreciate the reviewer's acceptance and recommendation of our work!